# OpenReview forum: "MultiBreak: A Scalable and Diverse Multi-turn Jailbreak Benchmark for Stress-testing LLM Safety"
_ICLR.cc/2026/Conference — Submitted to ICLR 2026_

### Official Review · Reviewer_hddK · 2025-10-29

**Soundness:** 3
**Presentation:** 3
**Contribution:** 2
**Rating:** 4
**Confidence:** 4

**Summary:**

The paper introduces MultiBreak, a benchmark for multi-turn jailbreaking attacks against LLMs. It first merges and cleans existing single- and multi-turn jailbreak datasets, to obtain a large and diverse pool of malicious intents and prompts. Then, via an active learning framework, fine-tunes an LLM to generate a large number of multi-turn conversation for jailbreaking the victim models. Finally, another LLM is used to refine the adversarial prompts to increase effectiveness. In the evaluation on open- and close-source LLMs, MultiBreak achieves higher success rate than existing benchmarks.

**Strengths:**

- The framework with the iterative addition of adversarial prompts and fine-tuning, followed by uncertainty-guided refinement, is reasonable and simple.

- The resulting adversarial prompts seem effective than the baselines according to the results in Table 2.

**Weaknesses:**

- The attack success rate is very different when evaluated with the two judges, and the difference is in opposite directions across models (i.e., it's not just that one judge is consistently more conservative). Since no evaluation of their alignment with human judgement is provided, it's unclear which one, if any, can be trusted.

- The reported success rate for different benchmarks is not obvious to compare: besides the issue with the automated judges (see above), different benchmarks have different unsafe goals (intents), so it's not clear that they're equally difficult. In general, I think there's a bit of confusion between the attack goals (what's the unsafe or harmful behavior the attacker wants to obtain) and method (in this case, how to generate the multi-turn prompts). The goals should stay fixed, and then different methods to achieve such goals could be compared. However, the paper collects both from previous benchmarks, and adds new ones. I think the proposed approach to generate multi-turn jailbreaks is promising, but should be compared to other methods on the same set of goals.

- Most of the conclusions in Sec. 4.4 are either known or expected.

**Questions:**

- The set of adversarial prompts seems fixed, and then, even if a model is robust to those, other reformulations could still work. Can the LLM-based generator (and rewriter) be used to generate new prompts in an on-line fashion, in case the fixed set is not effective?

---

> ### Author Response · Authors · 2025-11-21
>
> We deeply appreciate the review’s acknowledgement in our framework’s logistics. We address the questions and weaknesses below:
>
> > **Q1** Human judgement
>
> We include the table below for human judgement on MultiBreak across 5 models and 2 judges (**LG**: *Llama-Guard*, **GPT**: *GPT-4.1-mini*). We present agreement (**A**) [1], Cohen Kappa (**CK**) (agreement corrected for chance and robust to label imbalance) [2][3], and false positive rate (**FPR**).
>
> We observed that the agreement rate and Cohen Kappa with GPT judge on DeepSeek and Qwen3 is especially low, which reveals the reason for the inconsistency ASR reported in our Table 2. While the GPT judge shows a much **lower FPR**, it also gives lower agreement and Kappa. In contrast, LlamaGuard is more **conservative** and marks many harmless answers as unsafe, which increases agreement on harmful samples but also raises the false positive rate on harmless ones. This difference reflects a tradeoff in how the two judges behave. Therefore, these **judges are complementary to each other** and we cannot tell which judge is significantly better than the other one. We have added the results in Appendix A.5 in our submission.
> | Metric            | DeepSeek-7B | Qwen3-8B | Gemini-2.5-FL | GPT-4.1-mini | Llama-3.1-8B |
> |------------------|-------------|----------|----------------|--------------|--------------|
> | A(LG) ↑          | 0.772       | 0.792    | 0.779          | 0.792        | 0.926        |
> | CK(LG) ↑         | 0.145       | 0.124    | 0.526          | 0.338        | 0.827        |
> | FPR(LG) ↓        | 0.600       | 0.556    | 0.178          | 0.000        | 0.024        |
> | A(GPT) ↑         | 0.134       | 0.141    | 0.886          | 0.913        | 0.456        |
> | CK(GPT) ↑        | 0.008       | -0.002   | 0.738          | 0.536        | 0.147        |
> | FPR(GPT) ↓       | 0.000       | 0.111    | 0.133          | 0.182        | 0.024        |
>
> > **Q2** Confusion on attack goals
>
> We agree that each benchmark contains different goals, and we use the evaluation standard similar to paper [4]  for ASR evaluation. Papers would **use the same evaluation prompt** for LLM judges across the benchmarks, where each contains a different set of attack goals or intents. As the primary area of our paper is “datasets and benchmarks”, we consider the **collection of all intents to contribute to the diversity** of MultiBreak.
>
> > **Q3** Known or expected conclusions
>
> We would like to further include 3 insights from our proposed benchmark that we believe would provide useful and new analysis for multi-turn jailbreak research. Additionally, we would like to note reviewer BaJ7’s positive comment that *‘Fine-grained analysis reveals critical vulnerabilities’*.
>
> (1) **ASR of GPT-5 and GPT-OSS-20B**: We evaluate two state-of-the-art models (GPT-5 and GPT-OSS-20B) in the Appendix A.3.1 and show that **MultiBreak continues to reveal more failure cases** that do not appear under baseline multi-turn benchmarks. With up to 36.5% ASR, these SOTA models show lower vulnerability than the models in our main table, but they can still be broken in multi-turn settings.
>
> (2) **Category-level trends on defense methods**: From the coarse category-level breakdown for the two defense methods [5] and [6], we observe that the top-3 categories with the highest jailbreak proportion are *high-risk regulated advice, adult content, and harmful speech*, while the lowest proportions appear in categories related to *weapons, fraud, and cyber intrusion*.
>
> A notable result is the **strong reduction in cyber intrusion jailbreaks**: both defenses suppress this category to among the lowest-risk groups, even though cyber intrusion was originally one of the top-5 categories with the highest ASR without defenses (Figure 6.a in paper). This indicates that current **defending methods are more effective at blocking cyber-related attacks**, while categories such as high-risk advice continue to remain challenging.
>
> (3) **Refusal behavior across models**: We analyze **refusal patterns** across 5 victim models and find clear, model-specific behaviors that shape their vulnerability:
> - **DeepSeek-R1** sometimes *switch languages* inside one response or mix disclaimers with harmful strategies.
> - **Gemini 2.5-FL** consistently gives *structured explanations* for not answering, which keeps the conversation safe across turns.
> - **Llama-3.1** uses *short, direct refusals* but stops the dialogue early, preventing follow-up dialogues.
> - **GPT-4.1-mini and Qwen-3** rarely refuse and *continue the dialogue* even in risky contexts after warning or refusal, which might increase its exposure to multi-turn attacks.
>
> These refusal styles help explain the model-wise failure modes observed in our experiments. We have included examples in Appendix A.4.1.

---

> ### Author Response · Authors · 2025-11-21
>
> > **Q4** Generalizability of the pipeline
>
> Ablation study in appendix A.3.2 shows the results using new intents collected from dataset [7], where our generator can **effectively generate** harmful adversarial prompts **without finetuning the model**. The highest ASR@1 is 20.8% on the Llama3.1-8B-Instruct model while our **generation remains faithful to the input intents**.
>
> ----
> References:
>
> [1]: Judging LLM-as-a-Judge with MT-Bench and Chatbot Arena
>
> [2]: SORRY-Bench: Systematically Evaluating Large Language Model Safety Refusal
>
> [3]: A Coefficient of Agreement for Nominal Scales
>
> [4]: Reasoning-Augmented Conversation for Multi-Turn Jailbreak Attacks on Large Language Models
>
> [5]: X-Boundary: Establishing Exact Safety Boundary to Shield LLMs from Multi-Turn Jailbreaks without Compromising Usability
>
> [6]: Steering Dialogue Dynamics for Robustness against Multi-turn Jailbreaking Attacks
>
> [7]: The Multilingual Alignment Prism: Aligning Global and Local Preferences to Reduce Harm

---

> ### Author Response · Authors · 2025-11-25
> **Looking forward to more discussions.**
>
> Dear Reviewer hddK:
>
> As the author-reviewer discussion period has started for a few days, we will appreciate if you could check our response to your review comments soon. This way, if you have further questions and comments, we can still reply before the author-reviewer discussion period ends. If our response resolves your concerns, we kindly ask you to consider raising the rating of our work. Thank you very much for your time and efforts.
>
> Best regards,
>
> Authors of Submission 20915

---

> > ### Comment · Reviewer_hddK · 2025-11-26
> >
> > I thank the authors for the response and additional experiments.
> >
> > First, I think the low agreement or high FPR of the judges when compared to human evaluation is problematic. Also, I don't think that in the case having complementary automated judges is a strength, as it means that neither one can be trusted and evaluation is unreliable.
> >
> > Second, the fact that the paper is in the “datasets and benchmarks” area doesn't mean that any dataset or benchmark constitutes a positive contribution. Moreover, the paper explicitly claims the proposed dataset to be better since it achieves higher ASR (boldfacing in Table 2), which as mentioned in the review may not be indicative.
> >
> > Finally, the novel analyses mentioned in the rebuttal seem quite high-level observations, without actionable insights.
> >
> > Overall, I think in the current version the proposed benchmark does not have a solid design, and does not provide novel insights.

---

> ### Author Response · Authors · 2025-11-27
>
> Thank you very much for these thoughtful comments. We appreciate the opportunity to clarify these points.
>
> >On the judge's reliability concern.
>
> It is **widely and commonly observed** that automated judges have different agreements and FPR compared to human evaluation [1,2,3,4]. Our analysis in Section 4.4 illustrates the category-wise difference among judges, which motivated us to use multiple judges. This variation is precisely why we use multiple judges rather than relying on a single one. Here, we further include a new result with another judge, Qwen3 Guard 8B [5], the SOTA safety guardrail model published in October 2025, which was *even **after** the ICLR submission deadline*. The ASR result also demonstrates the high vulnerability of our dataset.
>
> | Method   | llama3.1-8b | gpt-4.1-mini | deepseek-7b | qwen3-8b |
> |----------|-------------|--------------|-------------|----------|
> | CoSafe   | 0.231       | 0.138        | 0.236       | 0.220    |
> | MHJ      | 0.460       | 0.493        | 0.317       | 0.569    |
> | SafeDial | 0.432       | 0.331        | 0.283       | 0.468    |
> | RedQueen | 0.106       | 0.141        | 0.287       | 0.310    |
> | **Ours** | **0.485**   | **0.650**    | **0.785**   | **0.752** |
>
> Moreover, regarding the quality of the dataset, we would like to highlight that qNt2 mentioned dataset is “carefully examined” and Lngo mentioned “the data generation pipeline is clear and makes sense to me, which is convincing of its high quality”.
>
> >On whether the benchmark is a positive contribution.
>
> We would like to re-emphasize our core motivation and contribution:
>
> 1) We **point out and correct the weaknesses and bottlenecks of existing jailbreak benchmarks**, and higher ASR is a *natural positive outcome* from our calibrations.
> We first show high redundancy in existing benchmarks, and then emphasize the diversity, coverage, and scale of the dataset, as shown in the coarse- and fine-grained breakdowns and the category distribution tables.
>
> 2) The **consistently high ASR** under **both automated judges** indicates that the dataset exposes failures across a wide range of categories. Additional analysis in Appendix A.3.1 further shows that our benchmark exposes vulnerability in SOTA models.
>
> >On the comment about actionable insights.
>
> Developed from the insights, we further proposed the following actionable steps:
>
> - **Multi-turn focused evaluation metric to capture intention escalation**: Our MultiBreak insight in Section 4.4 point 2 and Figure 6.b shows that many jailbreaks emerge from gradual, multi-turn drift, suggesting that future defenses should incorporate dialogue-level tracking, such as intent accumulation or turn-level risk scoring, instead of using single-turn classifiers.
> - **Defense mechanisms should consider per-category metrics**: The category-level breakdown in the additional insights provided shows that defenses sharply suppress cyber intrusion but remain weak on high-risk advice, adult content, and harmful speech. This motivates category-focused evaluation and targeted data augmentation or fine-tuning, rather than relying only on a single aggregated ASR.
> - **Refusal behavior as an indicator of vulnerability**: In Appendix A.4.1, we show that since some models continue the dialogue after refusing and then become more vulnerable, one mitigation approach is to encourage early termination or enforce a consistent refusal template at the system-prompt level.
> - **Jailbreaks tailored to SOTA model weaknesses**: MultiBreak uncovers failure cases for GPT-5 and GPT-OSS-20B that do not appear in baseline benchmarks, shown in Appendix A.3.1, which gives a clear next step: we can organize these harder cases into a stress-test subset for models with stronger safety alignment training.
> -----
> References:
>
> [1]: SORRY-Bench: Systematically Evaluating Large Language Model Safety Refusal
>
> [2]: CoSafe: Evaluating Large Language Model Safety in Multi-Turn Dialogue Coreference
>
> [3]: A STRONGREJECT for Empty Jailbreaks
>
> [4]: HarmBench: A Standardized Evaluation Framework for Automated Red Teaming and Robust Refusal
>
> [5]: Qwen3Guard Technical Report

---

### Official Review · Reviewer_qNt2 · 2025-10-30

**Soundness:** 4
**Presentation:** 4
**Contribution:** 2
**Rating:** 6
**Confidence:** 3

**Summary:**

This paper mainly proposes a new multi-turn jailbreak dataset called MultiBreak, which covers a larger sample size and a broader range of malicious intent. To ensure the high-quality of the data sample and increase the data size, the authors first collect and filter exsiting multi-turn and single-turn malicious samples, then use them to train specific generators to generate multi-turn jailbreak sampels. There are also three concerns to the generated data: the attack success rate, the faithfulness and uncertainty, which combine to ensure the high-quality of the generated data. The final dataset is used to the stress-testing of LLMs and once again demonstrates the insufficient defensive capabilities of LLMs against multi-turn dialogue attacks.

**Strengths:**

1. The samples in the dataset are carefully examined. For collected data, deduplication is performed based on semantic similarity, and filtering is conducted based on ASR. For generated data, filtering is performed based on ASR, intent consistency, and uncertainty (cross-model attack performance).
2. The ensemble approach is employed for ASR evaluation and multi-round jailbreak sample generation, which may increase the final quality of the data sample.
3. The writing is clear and easy to follow.

**Weaknesses:**

**1. There seems to be little novel scientific insight via the MultiBreak:**

Although the process of creating the dataset is highly rigorous, there seems to be little new finding enabled by MultiBreak. For example, the main claim in contribution 3 that 'LLMs resist overt harms but remain weak to subtle harms' has already been widely known. The conclusion and findings via MultiBreak are more like summaries of the known vulnerabilities in LLM safety.

**2. Lack of countermeasures to address the long-tail effect in the dataset:**
The lack of countermeasures to address the long-tail effect in the dataset may cause generators to overfit specific jailbreak attack topics during fine-tuning. This leads to generators gradually becoming capable of generating high-quality jailbreak attack prompts only for specific jailbreak topics. Combined with the existence of filtering mechanisms, this imbalance in sample quantities across different jailbreak topics may be further exacerbated. Therefore, the scale-up process may ultimately undermine the utility of the dataset.

**Questions:**

See the weakness above and:
1. This work is more like an **Engineering** work, but not **Research**. So I am uncertain whether it is suitable for publication as a research paper.

**Details Of Ethics Concerns:**

This paper constructs a new multi-turn jailbreak dataset but has Ethics Statement section.

---

> ### Author Response · Authors · 2025-11-21
>
> We deeply appreciate the review’s acknowledgement in the quality of MultiBreak. We address the questions and weaknesses below:
>
> > **Q1** Little new finding / Engineering-focused work
>
> We would like to further include 3 insights from our proposed benchmark that we believe would provide useful and new analysis for multi-turn jailbreak research.
>
> (1) **ASR of GPT-5 and GPT-OSS-20B**: We evaluate two state-of-the-art models (GPT-5 and GPT-OSS-20B) in the Appendix A.3.1 and show that **MultiBreak continues to reveal more failure cases** that do not appear under baseline multi-turn benchmarks. With up to 36.5% ASR, these SOTA models show lower vulnerability than the models in our main table, but they can still be broken in multi-turn settings.
>
> (2) **Category-level trends on defense methods**: From the coarse category-level breakdown for the two defense methods [1] and [2], we observe that the top-3 categories with the highest jailbreak proportion are *high-risk regulated advice, adult content, and harmful speech*, while the lowest proportions appear in categories related to *weapons, fraud, and cyber intrusion*.
>
> A notable result is the **strong reduction in cyber intrusion jailbreaks**: both defenses suppress this category to among the lowest-risk groups, even though cyber intrusion was originally one of the top-5 categories with the highest ASR without defenses (Figure 6.a in paper). This indicates that current **defending methods are more effective at blocking cyber-related attacks**, while categories such as high-risk advice continue to remain challenging.
>
> (3) **Refusal behavior across models**: We analyze **refusal patterns** across 5 victim models and find clear, model-specific behaviors that shape their vulnerability:
> - **DeepSeek-R1** sometimes *switch languages* inside one response or mix disclaimers with harmful strategies.
> - **Gemini 2.5-FL** consistently gives *structured explanations* for not answering, which keeps the conversation safe across turns.
> - **Llama-3.1** uses *short, direct refusals* but stops the dialogue early, preventing follow-up dialogues.
> - **GPT-4.1-mini and Qwen-3** rarely refuse and *continue the dialogue* even in risky contexts after warning or refusal, which might increase its exposure to multi-turn attacks.
>
> These refusal styles help explain the model-wise failure modes observed in our experiments. We have included examples in Appendix A.4.1.
>
> > **Q2** Long-tail effect in dataset
>
> We compare our intent topic distribution with our final curated dataset distribution in the below table and report the sorted absolute difference. It shows that the distribution in most coarse categories **did not shift much**. Especially, the originally long-tailed categories, such as *self-harm, weapons, and adult content*, remain at similar proportions in the final curated dataset. This indicates that the scale-up process of **our pipeline is able to generate prompts in different categories**.
>
> We notice larger changes in proportion in *fraud, cyber intrusion and high-rise categories*. The shifts we observe occur primarily in categories that are either *empirically harder* for current LLMs to jailbreak or are *less aggressively suppressed* by the target models’ safety training.
>
> This shows that the **distribution shift reflects category difficulty, not generator overfitting** since long-tail categories remain represented, minority categories do not vanish, and the overall ordering of category frequencies is preserved.
>
> | Category                                      | Original | Final | Abs. Diff |
> |-----------------------------------------------|----------|-------|-----------|
> | Self-Harm & Manipulative AI Risks             | 0.037    | 0.027 | 0.010     |
> | Weapons & Hazardous Substances                | 0.088    | 0.075 | 0.013     |
> | Privacy, Property & Public Order Offenses     | 0.145    | 0.132 | 0.013     |
> | Harmful Speech & Information Integrity        | 0.175    | 0.159 | 0.016     |
> | Sexual & Adult Content                        | 0.046    | 0.023 | 0.023     |
> | Violence & Exploitation                       | 0.125    | 0.087 | 0.038     |
> | Cyber Intrusion & Malware                     | 0.100    | 0.162 | 0.062     |
> | Fraud & Market Deception                      | 0.202    | 0.087 | 0.116     |
> | High-Risk Regulated Advice                    | 0.082    | 0.248 | 0.166     |
>
> ----
> References:
>
> [1]: X-Boundary: Establishing Exact Safety Boundary to Shield LLMs from Multi-Turn Jailbreaks without Compromising Usability
>
> [2]: Steering Dialogue Dynamics for Robustness against Multi-turn Jailbreaking Attacks

---

> > ### Comment · Reviewer_qNt2 · 2025-11-26
> >
> > Thank the authors for their response to my review comments.
> >
> > For my Q1, I see that the authors have concluded more detailed features about the multi-turn jailbreak, these could be a bonus, but there still seems to be no new vulnerability found by this proposed benchmark.
> >
> > For my Q2, thank you for clarifying that minority categories do not vanish during the generation. I know there may be a long-tail effect from the 'originally long-tailed categories', and that is the point that should be addressed, just maintaining the proportion may not be enough.
> >
> > For my Q3, there seems to be no defense. Though I may like to maintain my evaluation at this stage, I think I need to lower my confidence.

---

> > > ### Author Response · Authors · 2025-11-27
> > >
> > > Thank you very much for these thoughtful comments. We will address each point in detail below.
> > >
> > > > Q1.
> > >
> > > Thank you for the acknowledgement of the additional insights. We would like to briefly clarify the vulnerabilities and practical directions that MultiBreak uncovers but are not captured in prior benchmarks. As shown in Section 4.4 point 2 and Figure 6.b, the multi-turn setting *reveals vulnerabilities* that arise specifically from *intention escalation and dialogue drift*, which cannot be detected by single-turn classifiers and indicate **the need for multi-turn–aware evaluation metrics**. The per-category comparison in the additional insights further shows that *harm areas respond differently* under the same defense, suggesting that **category-aware evaluation or targeted data updates may be beneficial**. In Appendix A.4.1, we also observe that *refusal patterns can relate to later vulnerability*, and a **more consistent refusal strategy** may help mitigate this behavior. Finally, Appendix A.3.1 shows that MultiBreak *uncovers cases that emerge only in stronger models*, offering **stress-test examples** that may support future safety alignment efforts.
> > >
> > > > Q2.
> > >
> > > Thanks for further suggesting the “long-tail effect from the 'originally long-tailed categories'.” We acknowledge that this original long-tail distribution is *commonly shared* across all jailbreaking datasets. We are **actively running** experiments to try to *balance the original category distribution* before our iterative data generation, but during rebuttal, we may not have enough time to collect enough samples for minority categories. We *promise to include a new version of our dataset with a more balanced original distribution*.
> > >
> > > > Q3.
> > >
> > > Thank you for this question!
> > >
> > > First, we emphasize that in the LLM era, ***curating and synthesizing data is a genuine research discipline***, not merely “data engineering.” Data-centric AI now treats the design of training and evaluation corpora as a core scientific problem rather than an implementation detail. Many of the real gains in alignment, reasoning, and tool use come from new human and synthetic datasets and proprietary “data engines,” which are on par with advances in model and algorithm design [1, 2, 3].
> > >
> > > Second, our dataset synthesis effort is a critical and non-negotiable component of our scientific contributions and our actionable insights into LLM vulnerability to jailbreaking attacks. We **identify and correct key weaknesses and bottlenecks in existing jailbreak benchmarks**, including high redundancy, single-turn interactions, and low diversity, coverage, and scale. Our higher ASR is a natural consequence of these improvements. Without first diagnosing and calibrating these shortcomings, there would be ***no hope*** of revealing fine-grained vulnerabilities in LLMs or proposing meaningful defence strategies (as elaborated in our response to Q1 above and in Appendix A.4).
> > >
> > > -----
> > > References:
> > >
> > > [1]: Data-Centric Artificial Intelligence
> > >
> > > [2]: Why the Data Ocean Is Being Sectioned Off
> > >
> > > [3]: Training language models to follow instructions with human feedback

---

> > > > ### Comment · Reviewer_qNt2 · 2025-11-27
> > > >
> > > > I would like to thank the authors for their further clarification, and I have raised my confidence score.

---

> ### Author Response · Authors · 2025-11-25
> **Looking forward to more discussions.**
>
> Dear Reviewer qNt2:
>
> As the author-reviewer discussion period has started for a few days, we will appreciate if you could check our response to your review comments soon. This way, if you have further questions and comments, we can still reply before the author-reviewer discussion period ends. If our response resolves your concerns, we kindly ask you to consider raising the rating of our work. Thank you very much for your time and efforts.
>
> Best regards,
>
> Authors of Submission 20915

---

### Official Review · Reviewer_Lngo · 2025-11-01

**Soundness:** 3
**Presentation:** 3
**Contribution:** 2
**Rating:** 4
**Confidence:** 4

**Summary:**

The paper shows a benchmark and dataset for multi-turn jailbreak attacks, by unifying harmful intents and introducing an active learning pipeline for better adversarial prompts. Jailbreak generator is iteratively trained for stronger attacks guided by uncertainty. 7k prompts with 1.7k intents are proposed. Experiments show the results under the current multiple LLMs with the findings.

**Strengths:**

The dataset's 26 categories are comprehensive and fine-grained, spanning 9 courses. The data generation pipeline is clear and makes sense to me, which is convincing of its high quality. Most of the experiment analysis is impressive and comprehensive.

**Weaknesses:**

- During the iterative attack generator finetuning, once the new adversarial prompt is generated, it is unclear how the duplication can be checked again and if the diversity will be reduced.

- Experiments should also include some existing multi-turn defense methods [1,2] or naive safety post-training methods as baselines to show that the new benchmark is challenging enough for current defense methods.

- The experimental setup is not detailed, e.g., it is unclear what the temperature is for the @1 and @5 calculations, since it affects the number of trials significantly.

References:

[1] Hu et al. Steering Dialogue Dynamics for Robustness against Multi-turn Jailbreaking Attacks, 2025

[2] Lu et al. X-Boundary: Establishing Exact Safety Boundary to Shield LLMs from Multi-Turn Jailbreaks without Compromising Usability, 2025

**Questions:**

See above. I would be happy to raise my rating if all my concerns are addressed.

---

> ### Author Response · Authors · 2025-11-21
>
> We deeply appreciate the review’s acknowledgement in the diversity and quality of MultiBreak. We address the questions and weaknesses below:
>
> > **Q1** Duplication handling in pipeline
>
> At every finetuning iteration, we collect the harmful intents and their corresponding multi-turn adversarial prompts that consistently jailbreak many models. Once an intent has produced a strong adversarial prompt, we remove that intent from the remaining candidate pool **to avoid repeatedly generating similar attacks**. We have added the clarification in Section 3.3.2 Filtering in our submission.
>
> > **Q2** Comparison on defending methods
>
> We evaluate our benchmark with the two defending methods [1] and [2], and report the results in the table below. As shown, the ASR@1 decreases when these defenses are applied, indicating that they are effective in filtering or blocking attacks, especially in cyber intrusion, fraud, and weapons categories.
>
> However, the ASR remains noticeably above zero on all tested models, which shows that the **current defending methods still cannot fully prevent multi-turn jailbreaks**. We note that X-Boundary cannot be evaluated on closed-source models because its method requires an additional finetuned adapter, and only open-source checkpoints are publicly available.
>
> | Method      | gpt-4.1-mini           | llama3.1-8b|
> |-------------|-------------------------|------------------------|
> | X-Boundary  | N/A (closed-source)     | 0.181                 |
> | NBF-LLM     | 0.194                   | 0.240                 |
> | **Ours**    | 0.710               | 0.621             |
>
> > **Q3** Experiment setup
>
> During data curation, we set the victim model’s temperature to 0, and during evaluation, we use temperature 1. To verify that setting temperature to1 does not substantially affect the reported ASR, we conducted an additional experiment on a subsample using temperature 0 during evaluation. The results (in the table below) differ by only 1 to 6 percentage points across ASR@1 to ASR@10, confirming that the **evaluation is stable under different temperature settings**. We have added a description for victim model temperature in Section 4.1 in the submission.
>
> | Metric   | gpt-4.1-mini | llama3.1-8b|
> |----------|--------------|-----------------------|
> | ASR@1    | 0.691        | 0.588                 |
> | ASR@5    | 0.785        | 0.865                 |
> | ASR@10   | 0.832        | 0.899                 |
>
> -----
> References:
>
> [1]: X-Boundary: Establishing Exact Safety Boundary to Shield LLMs from Multi-Turn Jailbreaks without Compromising Usability
>
> [2]: Steering Dialogue Dynamics for Robustness against Multi-turn Jailbreaking Attacks

---

> ### Author Response · Authors · 2025-11-25
> **Looking forward to more discussions.**
>
> Dear Reviewer Lngo:
>
> As the author-reviewer discussion period has started for a few days, we will appreciate if you could check our response to your review comments soon. This way, if you have further questions and comments, we can still reply before the author-reviewer discussion period ends. If our response resolves your concerns, we kindly ask you to consider raising the rating of our work. Thank you very much for your time and efforts.
>
> Best regards,
>
> Authors of Submission 20915

---

> > ### Comment · Reviewer_Lngo · 2025-11-26
> >
> > Thank you for carrying out the rebuttal. Most of the concerns have been dismissed, but the additional experiment results during the rebuttal should be updated in the revised version of the paper, where I cannot find many revisions highlighted currently, even in the appendix. I am willing to raise my score if the paper is well-revised.

---

> > > ### Author Response · Authors · 2025-11-27
> > >
> > > Thank you for the acknowledgement. We have included the experiment results of both defending methods and ablation studies on temperature into Appendix A.3.3 and A.3.4, respectively.

---

### Official Review · Reviewer_BaJ7 · 2025-11-01

**Soundness:** 2
**Presentation:** 3
**Contribution:** 2
**Rating:** 4
**Confidence:** 3

**Summary:**

This paper presents MultiBreak, a multi-turn jailbreak benchmark containing 7,152 adversarial prompts across 1,724 unique harmful intents—significantly larger and more diverse than existing datasets. The authors employ an active learning pipeline that iteratively fine-tunes attack generators and applies uncertainty-guided rewriting to scale up high-quality adversarial examples. MultiBreak achieves substantially higher attack success rates than prior benchmarks (up to 54.1% improvement on DeepSeek-R1-7B), and reveals that LLMs are more vulnerable to subtle harms (e.g., unsafe medical advice) than overt harms (e.g., hate speech), with framing-based attacks proving especially effective at bypassing safety guardrails.

**Strengths:**

With 7,152 prompts spanning 1,724 unique intents across 26 fine-grained categories, MultiBreak is significantly larger and more diverse  than prior multi-turn jailbreak benchmarks.

Fine-grained analysis reveals critical vulnerabilities—subtle harms (unsafe medical advice) are easier to bypass than overt harms (hate speech), framing-based attacks are most effective, and conversation strategy matters more than length.

**Weaknesses:**

The generator pre-scripts all conversation turns (Turn 1, 2, 3, etc.) upfront without seeing actual victim responses, so Turn 2 cannot reference what the victim (which may vary) actually said in Response 1, breaking natural conversational flow.

For a safety benchmark, lack of human judgment on whether attacks are genuinely harmful (vs. triggering false positives) is a critical gap.

Some of the previous multi-turn jailbreaking methods, such as X-Teaming (https://arxiv.org/abs/2504.13203
) and ActorAttack (https://arxiv.org/abs/2410.10700
), extend single-turn benchmarks (like HarmBench) into multi-turn. However, discussion/comparison with such methods is missing.

**Questions:**

Did the authors analyze whether their pre-scripted multi-turn attacks maintain effectiveness and contextual coherence when evaluated on completely different model families (e.g., Claude/Grok series) beyond the tested victims?

---

> ### Author Response · Authors · 2025-11-20
>
> We deeply appreciate the reviewer's acknowledgement in the scale and diversity of MultiBreak and the fine-grained analysis. We address the questions and weaknesses below:
> > **Q1** Pre-scripted conversation flow
>
> In the table below, we compare embedding similarity (ES) and contextual coherence (Co.) between our benchmark and two representative baselines: [1] (template-based multi-turn benchmark) and [4] (on-the-fly attack generation). For ES, we compute the average cosine similarity between **each adjacent turn pair** in the conversation (e.g., prompt₁ ↔ response₁, response₁ ↔ prompt₂, etc.) using the “sentence_transformer/all-MiniLM-L6-v2” model. For Co., we ask GPT-4.1-mini to rate **each full conversation** from 1 (unnatural flow) to 5 (very natural flow), then normalize scores to [0, 1].
>
> Our benchmark shows **higher ES and comparable Co.** on both victim models (llama3.1-8b-instruct and gpt-4.1-mini), indicating that our multi-turn conversations form coherent turn-to-turn transitions rather than disconnected template stitching.
>
> We also note that prior benchmarks [1, 2] similarly use pre-scripted conversations, whereas [3, 5] generate interactions on-the-fly to specific victim models. However, interactive generations are **difficult to scale**, with [3] and [5] containing only 537 and 2,037 conversations respectively. We agree that extending our benchmark with interactive generation could be **a valuable next step**.
> | Method       | ES↑ (gpt-4.1-mini) | Co.↑ (gpt-4.1-mini) | ES↑ (llama-3.1-8b) | Co.↑ (llama-3.1-8b) |
> |--------------|-------------------|---------------------|--------------------|---------------------|
> | ActorAttack  | 0.527             | 0.879               | 0.491              | 0.756               |
> | RedQueen     | 0.355             | 0.995               | 0.289              | 0.784              |
> | **Ours**     | 0.602         | 0.947           | 0.626          | 0.706           |
>
> > **Q2** Human judgement
>
> We include the table below for human judgement on MultiBreak across 5 models and 2 judges (LG: Llama-Guard, GPT: GPT-4.1-mini). We present agreement (A) [6], Cohen Kappa (CK) (agreement corrected for chance and robust to label imbalance) [7][8], and false positive rate (FPR).
>
> We observed that the agreement rate and Cohen Kappa with GPT judge on DeepSeek and Qwen3 is especially low, which reveals the reason for the inconsistency ASR reported in our Table 2. While the GPT judge shows a much **lower FPR**, it also gives lower agreement and Kappa. In contrast, LlamaGuard is more **conservative** and marks many harmless answers as unsafe, which increases agreement on harmful samples but also raises the false positive rate on harmless ones. This difference reflects a tradeoff in how the two judges behave. Therefore, these **judges are complementary to each other** and we cannot tell which judge is significantly better than the other one. We have added the results in Appendix A.5 in our submission.
> | Metric            | DeepSeek-7B | Qwen3-8B | Gemini-2.5-FL | GPT-4.1-mini | Llama-3.1-8B |
> |------------------|-------------|----------|----------------|--------------|--------------|
> | A(LG) ↑          | 0.772       | 0.792    | 0.779          | 0.792        | 0.926        |
> | CK(LG) ↑         | 0.145       | 0.124    | 0.526          | 0.338        | 0.827        |
> | FPR(LG) ↓        | 0.600       | 0.556    | 0.178          | 0.000        | 0.024        |
> | A(GPT) ↑         | 0.134       | 0.141    | 0.886          | 0.913        | 0.456        |
> | CK(GPT) ↑        | 0.008       | -0.002   | 0.738          | 0.536        | 0.147        |
> | FPR(GPT) ↓       | 0.000       | 0.111    | 0.133          | 0.182        | 0.024        |
>
> > **Q3** Attack method comparison
>
> In the table below, we report the ASR@1 for X-Teaming [9] and ActorAttack [4], all evaluated on our single-turn intents, extended to multi-turn attacks using the public code released by each paper. This allows all three methods to be compared under the same data and the same judge (Llama-Guard).
>
> Under these conditions, both X-Teaming and ActorAttack obtain lower ASR@1 than ours, which shows that **our benchmark remains strong** when evaluated under the same setup. We note that both X-Teaming and ActorAttack generate multiple attack strategies per goal using closed-source models, which is **more costly** than our method. Because we evaluate ASR@1, we set the attack budget to a single attempt per intent for all methods. The original papers use larger budgets (e.g., multiple retries or multiple generated strategies), but those settings are not compatible with ASR@1.
>
>  | Method       | gpt-4.1-mini | llama3.1-8b|
> |--------------|--------------|-----------------------|
> | X-Teaming    | 0.499        | 0.301                 |
> | ActorAttack  | 0.380        | 0.412                 |
> | **Ours**     | 0.710    | 0.621             |

---

> ### Author Response · Authors · 2025-11-20
>
> > **Q4** Analysis on other model families
>
> We evaluate our benchmark on two non-overlapping families: grok-4-fast and claude-Haiku-3.5. We compare our benchmark against a baseline benchmark RedQueen [1], using (1) Embedding Similarity (ES) for turn-to-turn semantic alignment, (2) Contextual Coherence (Co.) rated by GPT-4.1-mini for conversation flow, and (3) ASR at one pass for attack success.
>
> Table below shows that our benchmark has higher embedding similarity and comparable coherence to RedQueen, indicating that the **scripted turns remain natural**. Meanwhile, our benchmark achieves much higher ASR on both models, confirming that the multi-turn prompts **still transfer as strong jailbreaks** beyond the original victims.
>
> These results confirm that our benchmark can **effectively analyze beyond the models used during curation**, maintaining both effectiveness and natural multi-turn behavior when evaluated on Claude and Grok.
>
> | Metric | grok-4-fast (RedQueen) | grok-4-fast (Ours) | claude-haiku-3.5 (RedQueen) | claude-haiku-3.5 (Ours) |
> |--------|-------------------------|---------------------|------------------------------|---------------------------|
> | ES     | 0.342                  | 0.638               | 0.340                        | 0.582                     |
> | Co.    | 0.942                  | 0.940               | 0.969                        | 0.953                     |
> | ASR@1    | 0.004                  | 0.232               | 0.000                        | 0.110                     |
> ----
> References:
>
> [1]: RED QUEEN: Safeguarding Large Language Models against Concealed Multi-Turn Jailbreaking
>
> [2]: CoSafe: Evaluating Large Language Model Safety in Multi-Turn Dialogue Coreference
>
> [3]: LLM Defenses Are Not Robust to Multi-Turn Human Jailbreaks Yet
>
> [4]: Derail Yourself: Multi-turn LLM Jailbreak Attack through self-discovered clues
>
> [5]: SafeDialBench: A Fine-Grained Safety Benchmark for Large Language Models in Multi-Turn Dialogues with Diverse Jailbreak Attacks
>
> [6]: Judging LLM-as-a-Judge with MT-Bench and Chatbot Arena
>
> [7]: SORRY-Bench: Systematically Evaluating Large Language Model Safety Refusal
>
> [8]: A Coefficient of Agreement for Nominal Scales
>
> [9]: X-Teaming: Multi-Turn Jailbreaks and Defenses with Adaptive Multi-Agents

---

> ### Author Response · Authors · 2025-11-25
> **Looking forward to more discussions.**
>
> Dear Reviewer BaJ7:
>
> As the author-reviewer discussion period has started for a few days, we will appreciate if you could check our response to your review comments soon. This way, if you have further questions and comments, we can still reply before the author-reviewer discussion period ends. If our response resolves your concerns, we kindly ask you to consider raising the rating of our work. Thank you very much for your time and efforts.
>
> Best regards,
>
> Authors of Submission 20915

---

> ### Author Response · Authors · 2025-11-27
> **Anticipating Further Discussions**
>
> Dear Reviewer BaJ7,
>
> We would like to kindly remind you that the author-reviewer discussion period has started for several days now and will end soon. We would greatly appreciate it if you could review our responses to your initial comments at your earliest convenience. This will enable us to address any additional queries or feedback you might have before the discussion period ends.
>
> Should our responses sufficiently address your concerns, we respectfully request that you consider raising the rating of our work.
>
> Thank you very much for your attention, time, and efforts!
>
> Best regards,
>
> Authors of Submission 20915

---

### Author Response · Authors · 2025-11-21

We sincerely thank all reviewers for their constructive and thoughtful feedback. We especially appreciate *BaJ7, qNt2, and Lngo*’s **acknowledgment of the quality** of our benchmark, as well as *hddK and qNt2*’s comments on the **effectiveness of our pipeline**. We are also grateful for the high-quality comments from each reviewer, which helped us refine the paper and make it more comprehensive.

We have updated our submission (revised text shown in blue) to incorporate the following improvements:
- clarification of the victim model temperature setup in Section 4.1
- human judgement results in Appendix A.5
- explanation of duplication handling in Section 3.3.2
- expanded analysis of refusal patterns in Appendix A.4.1

While we have addressed each concern in detail in the individual responses, we would like to offer one *general clarification*. Unless otherwise stated, we use gpt-4.1-mini (closed-source representative) and llama-3.1-8b-instruct (open-source representative) as our victim models in all attack and defense experiments. This ensures that the **results remain consistent with the main comparisons** presented in Table 2 of our paper.

---

### Author Response · Authors · 2025-11-24
**Looking forward to more discussions.**

Dear Reviewers:

As the author-reviewer discussion period has started for a few days, we will appreciate if you could check our response to your review comments soon. This way, if you have further questions and comments, we can still reply before the author-reviewer discussion period ends. If our response resolves your concerns, we kindly ask you to consider raising the rating of our work. Thank you very much for your time and efforts.

Best regards,

Authors of Submission 20915

---

### Author Response · Authors · 2025-11-29
**Rebuttal Summary**

Dear PC, AC, and reviewers:

Since further public discussions are no longer allowed, we would like to post a rebuttal summary here to note all the revisions and updates made during this rebuttal period. We sincerely appreciate all the reviewers for their thoughtful suggestions that help improve the paper quality. Below, we list the main questions raised by the reviewers, and explain how we comprehensively addressed all questions. Specifically, before the revert, reviewer qNt2 promised to increase the confidence score, and reviewer Lngo has increased the score from 4 to 6. Therefore, before the revert, our final scores were:
- Reviewer BaJ7: 4
- Reviewer Lngo: 6
- Reviewer qNt2: 6
- Reviewer hddK: 4

**Main refinements:**

> Comparison experiment on more victim models, attack methods, and defense methods.

We conduct experiments on 2 new model families, Grok and Claude, to verify that MultiBreak shows higher vulnerability on other model families than the baseline datasets [(BaJ7 Q4)](https://openreview.net/forum?id=uJgfj5EJ2W&noteId=ab6icoDfDW). Further, we evaluate 2 attack methods, X-Teaming and ActorAttack, under the same data and same budget, which shows that our benchmark remains strong while the other attack methods are more costly due to the closed-source model used for generation [(BaJ7 Q3)](https://openreview.net/forum?id=uJgfj5EJ2W&noteId=m9rNIKP63b). We also test our dataset on 2 SOTA defense methods, X-Boundary and NBF-LLM, to demonstrate that current methods still cannot fully prevent multi-turn jailbreaks [(Lngo Q2)](https://openreview.net/forum?id=uJgfj5EJ2W&noteId=lpqLoml2JD).

> Human evaluation and judge reliability.

We include human judgement on MultiBreak across 5 models and 2 judges and show that the GPT judge has lower FPR while Llama-Guard has higher agreement rate and Cohen Kappa most of the time. We believe judges are complementary to each other [(BaJ7 Q2)](https://openreview.net/forum?id=uJgfj5EJ2W&noteId=m9rNIKP63b). To further demonstrate judge reliability, we use another automated judge, Qwen3 Guard 8B published in Oct. 2025, to show that our benchmark remains highly vulnerable compared to all baselines [(hddK 2nd round Q1)](https://openreview.net/forum?id=uJgfj5EJ2W&noteId=stjaFBZVi3).

> Additional insights with actionable steps. [(qNt2 Q1)](https://openreview.net/forum?id=uJgfj5EJ2W&noteId=p6hyqHk3mS) [(hddK 2nd round Q3)](https://openreview.net/forum?id=uJgfj5EJ2W&noteId=stjaFBZVi3)

As shown in Section 4.4 point 2 and Figure 6.b, the *multi-turn setting reveals vulnerabilities* that arise specifically from *intention escalation and dialogue drift*, which cannot be detected by single-turn classifiers and indicate ***the need for multi-turn–aware evaluation metrics***.
The per-category ASR comparison on defending methods shows that *harm areas respond differently under the same defense*, suggesting that ***category-aware evaluation*** or targeted data fine-tuning may be beneficial.
In Appendix A.4.1, we also observe that *refusal patterns of LLMs can relate to later vulnerability*, and a more ***consistent refusal strategy*** may help mitigate this behavior.
Finally, Appendix A.3.1 shows that MultiBreak *uncovers cases that emerge only in stronger SOTA models*, GPT-5 and GPT-OSS-20B, and offering such ***stress-test examples*** would support future safety alignment efforts.

**For additional explanations or justifications, we direct to the rebuttal and discussions with reviewers:**
- Proof of natural conversation flow for pre-scripted generation: [BaJ7 Q1](https://openreview.net/forum?id=uJgfj5EJ2W&noteId=m9rNIKP63b)
- Proof for pipeline generalizability: [hddK Q4](https://openreview.net/forum?id=uJgfj5EJ2W&noteId=2EqAWInJED)
- Long-tail effect of dataset: [qNt2 Q2](https://openreview.net/forum?id=uJgfj5EJ2W&noteId=p6hyqHk3mS) and [further reply](https://openreview.net/forum?id=uJgfj5EJ2W&noteId=55Kfy974qf)
- Pipeline and evaluation detail justification: [Lngo Q1 and Q3](https://openreview.net/forum?id=uJgfj5EJ2W&noteId=lpqLoml2JD)
- Evaluation fairness to other baselines: [hddK Q2](https://openreview.net/forum?id=uJgfj5EJ2W&noteId=ipb0x6FDIq) and [further reply](https://openreview.net/forum?id=uJgfj5EJ2W&noteId=stjaFBZVi3
)

-----
In general, MultiBreak points out that existing multi-turn jailbreak benchmarks **suffer from redundancy, limited diversity, and narrow topic coverage**. Our benchmark addresses these limits by de-duplicating a broad set of harmful intents and by providing wider coarse- and fine-grained topic coverage supported by detailed category distributions. The resulting dataset exposes **high and persistent vulnerabilities** across victim models and demonstrates, in detail, the **prompt categories and attack methods** that remain highly vulnerable in multi-turn settings.

---

### Meta-Review · Area_Chair_utsV · 2026-01-07

**Summary:**

This paper presents MULTIBREAK, a scalable framework and benchmark designed to evaluate the safety of LLMs against multi-turn jailbreak attacks. The authors utilize an automated pipeline to generate multi-turn adversarial conversation scripts across various harmful intent categories, demonstrating that multi-turn interactions significantly increase attack success rates compared to single-turn baselines.

A major limitation is that the generator pre-scripts all conversation turns upfront without observing the actual responses from the victim model. Besides, the work is perceived by reviewers more as an engineering contribution than a scientific research paper.

**Reviewer Scores:**

No

---

### Decision · Program_Chairs · 2026-01-26

Reject